# Combined Subciliary/Transantral Approach for Reconstruction of Orbital Floor Fracture

**Norihiko Narita \*, Yumi Ito, Yukinori Kato, Yukihiro Kimura, Yoshimasa Imoto, Kazuhiro Ogi, Masayuki Okamoto, Tetsuji Takabayashi and Shigeharu Fujieda**

Department of Otorhinolaryngology, Head and Neck Surgery, Faculty of Medical Sciences, University of Fukui, 23 Shimoaizuki, Matsuoka, Fukui 910-1193, Japan; yumit@u-fukui.ac.jp (Y.I.); ykato@u-fukui.ac.jp (Y.K.); kimyuki@u-fukui.ac.jp (Y.K.); yimoto@u-fukui.ac.jp (Y.I.); ogikazu@u-fukui.ac.jp (K.O.); masayu@u-fukui.ac.jp (M.O.); tetsuji@u-fukui.ac.jp (T.T.); sfujieda@u-fukui.ac.jp (S.F.)
\* Correspondence: norihiko@u-fukui.ac.jp; Tel.: +81-(776)-61-8407; Fax: +81-(776)-61-8118

**Abstract:** Orbital floor fracture, especially with constriction of orbital soft tissue, should be reconstructed surgically. Although various approaches to treat the orbital floor have been reported, procedures have not been unified among hospitals or surgeons. Since 2009, we have adopted a procedure combining a transorbital approach via subciliary incision with a transantral approach through upper gingival incision. The combined approach compensates for the shortcomings of each approach, leading to successful reconstruction. It is applicable safely for trapdoor fracture of the orbital floor in children, which more frequently constricts orbital soft tissue and which leaves permanent diplopia. This report retrospectively assessed clinical preoperative findings and postoperative outcomes of patients who received reconstruction of orbital floor fracture with the combined approach in our department from August 2009 through March 2021. Data of 21 patients with orbital floor fracture were analyzed, only one (4.8%) of whom had postoperative diplopia. Specifically, we describe children with trapdoor fracture treated with the combined approach, resulting in complete recovery. The combined approach stands as an excellent procedure for reconstruction of orbital floor fracture in adults and even in children.

**Keywords:** blowout fracture; combined approach; orbital floor; reconstruction; subciliary incision; trapdoor fracture

## 1. Introduction

Orbital floor fracture is distinguished roughly into two groups, including open-type blowout fracture (BOF) and trapdoor fracture (TF), which is a so-called white-eyed BOF [1]. BOF caused by the breakdown of the orbital floor to various degrees can engender dislocation of orbital soft tissues such as fat tissue and extraocular muscle. In contrast, TF, which is more frequent in younger people, is induced by a narrower fracture or a linear fracture, resulting in extraocular muscle constriction [2]. Although orbital floor fracture of both types can cause diplopia, it is more severe in cases of TF. In addition, patients with TF can be seized by nausea, vomiting, or bradycardia derived from oculocardiac reflex [2]. Constriction of the inferior rectus muscle without early surgical reconstruction leads to permanent diplopia. Various surgical approaches, including transantral, transorbital, and endonasal endoscopic approaches, or a combination of them, have been performed for orbital floor reconstruction. We have used the combined surgical procedure with a subciliary transorbital and transantral approach using an endoscope for BOF and TF reconstruction since August 2009. The combined approach can be followed more safely and more effectively than single approaches for both BOF and TF reconstruction. This report, which retrospectively assessed clinical outcomes of the combined approach, describes the combined approach for TF reconstruction in children.

## 2. Materials and Methods

Based on University of Fukui Hospital medical records, clinical data of patients who underwent operation with the combined approach for BOF or TF between August 2009 and March 2021 were collected: age, gender, symptoms, fracture type, interval between injury and operation, operation time, follow up duration, and postoperative complication. Furthermore, the maximum fracture length was measured using preoperative CT images in the mediolateral (ML) direction and in the anteroposterior (AP) direction. Patients with multiple maxillofacial bone fractures or insufficient follow up were excluded. This study was approved by the ethical board of the University of Fukui (20210002C).

## 3. Results

Data of 21 patients (13 male, 8 female) were examined for this study, among those data, 18 BOF and 3 TF cases were identified. All cases were seized by at least diplopia before operation. Table 1 presents the preoperative findings and postoperative outcomes. The median age was 25 years old. The median fracture lengths were 14 mm (ML) and 17 mm (AP). The median intervals between injury and operation and the median operation time were, respectively, 10 days and 149 min. The reconstructed orbital floor was upheld with a balloon catheter. No implant was used for any case, as shown in case presentations. No esthetic complication or enophthalmos was found in any patient after surgery. Only one patient (4.8%) showed postoperative diplopia on upgaze, resulting in a complete recovery ratio of 95.2%.

**Table 1.** Preoperative and postoperative findings from patient data.

| Case Number | Age (Years) | Gender | Preoperative Symptom | Fracture Type | ML (mm) | AP (mm) | Interval (Days) | Operation Time (min) | Follow Up Duration (Months) | Postoperative Symptom |
|---|---|---|---|---|---|---|---|---|---|---|
| 1 | 18 | F | Diplopia on up- and down-gaze | BOF | 16 | 15 | 4 | 276 | 6 | None |
| 2 | 24 | M | Diplopia on down-gaze | BOF | 20 | 18 | 9 | 170 | 12 | None |
| 3 | 54 | F | Diplopia on up-gaze | BOF | 15 | 24 | 6 | 165 | 4 | None |
| 4 | 31 | M | Diplopia on up-gaze | BOF | 21 | 21 | 10 | 177 | 12 | None |
| 5 | 63 | M | Diplopia on up-gaze | BOF | 19 | 23 | 8 | 140 | 6 | None |
| 6 | 25 | F | Diplopia on up-gaze | BOF | 14 | 15 | 15 | 215 | 10 | None |
| 7 | 21 | M | Diplopia on up-gaze | BOF | 12 | 17 | 11 | 182 | 12 | None |
| 8 | 18 | M | Diplopia on up-gaze | BOF | 11 | 20 | 13 | 149 | 3 | None |
| 9 | 13 | F | Diplopia on up-gaze, nausea, pain on upgaze | TF | 7 | 6 | 0 | 164 | 5 | None |
| 10 | 8 | M | Diplopia on up-gaze vomitting, pain on upgaze | TF | 7 | 9 | 1 | 194 | 6 | None |
| 11 | 14 | M | Diplopia on up-gaze, vomitting, pain on upgaze | TF | 7 | 4 | 0 | 149 | 2 | None |
| 12 | 28 | M | Diplopia on up-gaze | BOF | 18 | 27 | 10 | 148 | 1 | None |
| 13 | 19 | F | Diplopia on up-gaze | BOF | 19 | 15 | 10 | 128 | 3 | None |
| 14 | 36 | F | Diplopia on down-gaze | BOF | 11 | 20 | 9 | 116 | 2 | None |
| 15 | 86 | M | Diplopia on up- and down-gaze | BOF | 14 | 21 | 18 | 148 | 11 | None |
| 16 | 39 | M | Diplopia on up- and down-gaze | BOF | 12 | 17 | 10 | 164 | 3 | None |
| 17 | 41 | F | Diplopia on up- and down-gaze | BOF | 14 | 16 | 1 | 123 | 3 | Diplopia on up-gaze |
| 18 | 36 | M | Diplopia on up-gaze | BOF | 15 | 24 | 13 | 116 | 2 | None |
| 19 | 22 | M | Diplopia on up-gaze | BOF | 14 | 8 | 11 | 122 | 1 | None |
| 20 | 25 | F | Diplopia on down-gaze | BOF | 13 | 15 | 9 | 114 | 1 | None |
| 21 | 69 | M | Diplopia on up-gaze | BOF | 15 | 31 | 16 | 139 | 2 | None |
| median | 25 | | | | 14 | 17 | 10 | 149 | 3 | |

BOF, blowout fracture; TF, trapdoor fracture; ML, maximum mediolateral length of fracture; AP, maximum anteroposterior length of fracture; Interval, interval between injury and operation.

*Case Presentation of TF Reconstruction*
Case 9

A 13-year-old girl injured her left eye by physical contact during a basketball game. Continuous ocular pain and vomiting led her to visit our hospital. Her left eyeball was locked completely (Figure 1), with accompanying violent pain and nausea in looking upward. A CT scan indicated orbital floor fracture on her left side (Figure 2A,B), in which the inferior rectus muscle might have been constricted. The CT scan and the clinical symptoms suggested the necessity of TF for immediate release of constriction of orbital soft tissue. The operation was performed on the day she was injured. We first released the periosteum of the orbital floor using a transorbital approach through subciliary incision. After the comprehensive release of periosteum around the fracture, the orbital soft tissues, including fat and muscle, were lifted up carefully. However, constriction prevented complete restoration (Figure 3A). Next, the transantral approach was conducted through the upper gingival incision, compatible with the conventional Caldwell–Luc procedure. Pieces of fractured orbital bone were observed clearly and were removed to release constriction of orbital soft tissue via the left maxillary sinus using an endoscope (Figure 3B). The dislocated orbital soft tissue was restored entirely. The orbital floor was upheld with a balloon catheter inserted through the inferior meatal nasoantral window. We used no implant on the orbital floor because the posterior ledge of the fracture was not observed clearly through a transorbital approach and because an inadequate implant can restrict eye movement. The anterior wall bone of the maxillary sinus was put back to its original place. Nausea and pain in her left eye were cured immediately by the operation. The balloon catheter was removed ten days after the reconstruction, leading eventually to complete improvement of eye movement and diplopia (Figure 4). The subciliary wound did not stand out esthetically. A CT scan one year after the operation showed the reconstructed orbital floor and clear sinuses of the left side (Figure 5A,B).

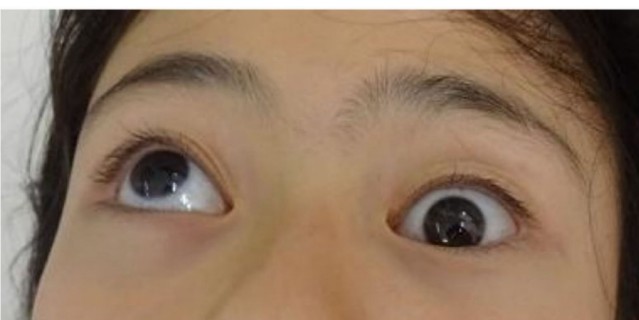

**Figure 1.** Case 9, eye movement when looking upward before reconstruction. The left eye is locked and impossible to upturn.

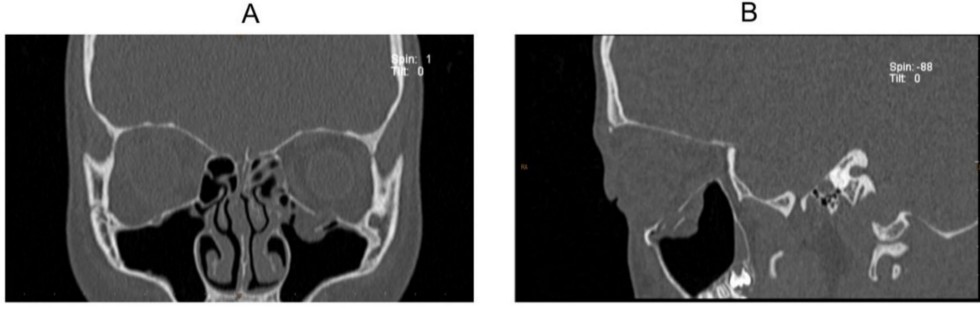

**Figure 2.** CT scan images of Case 9 before reconstruction: (**A**) coronal image and (**B**) sagittal image.

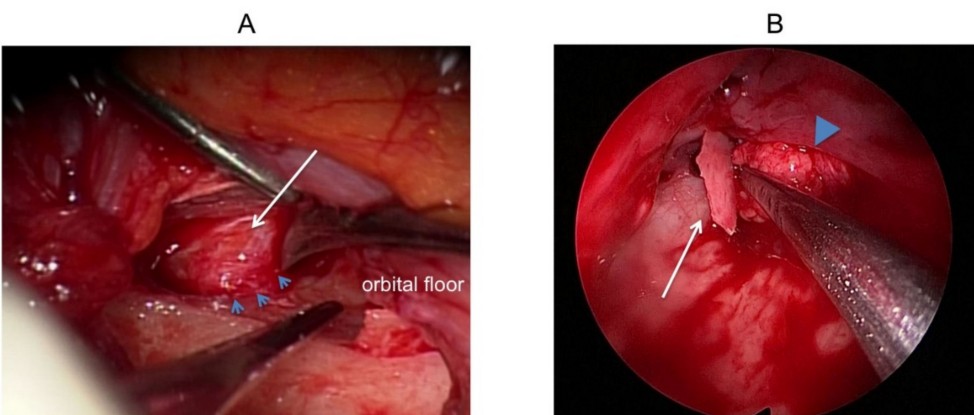

**Figure 3.** Findings in the operation of Case 9. (**A**) Transorbital approach using microscope. The orbital soft tissue is entrapped in the TF preventing complete restoration (arrow). The fracture line of the orbital floor is observed (arrowhead). (**B**) Transantral approach using endoscope. The constriction was released, removing fractured bone. The arrow indicates broken bone of the orbital floor. Orbital tissue is being pushed upward into the orbit (arrowhead).

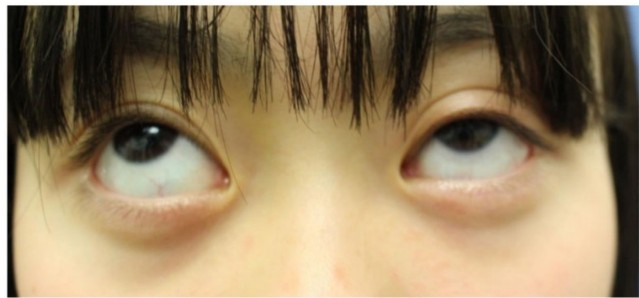

**Figure 4.** Case 9, eye movement when looking upward after reconstruction. The left eye can upturn normally. The subciliary wound does not stand out.

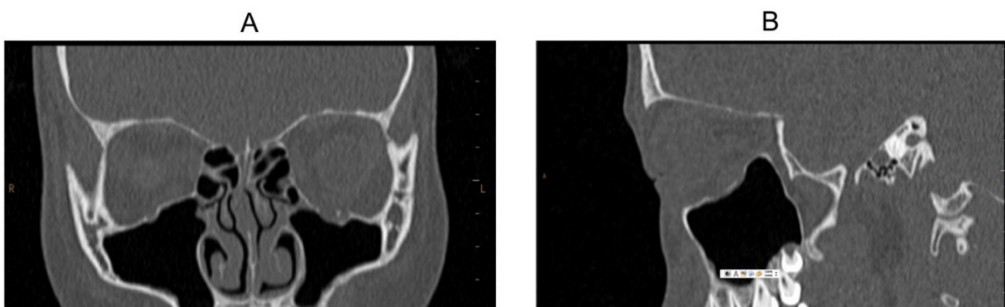

**Figure 5.** CT scan images of Case 9 after reconstruction: (**A**) coronal image and (**B**) sagittal image. Constriction of the orbital floor is released. The left maxillary sinus showed good aeration.

Case 10

An 8-year-old boy was injured in his left eye by a collision with his friend. He visited an ophthalmologist's clinic with vomiting and severe pain in his left eye. The next day, he was introduced to our hospital. He was hospitalized for an emergency operation. His left eye movement was limited upward (Figure 6A). The CT scan showed entrapment of the orbital soft tissue in the TF (Figure 6B). The left orbital medial wall was also fractured, not causing impaired horizontal ocular movement. The operation was performed in the same mode used for Case 9 with a combined approach for TF of the orbital floor. Pain and vomiting were improved completely through the operation. By removing bone pieces, lifting the orbital contents upward, and upholding the orbital floor with a balloon catheter, his eye movement recovered entirely without diplopia. The subciliary wound left no

esthetic problem (Figure 6C). Subsequently, a CT scan indicated the restored left orbital floor without implant (Figure 6D).

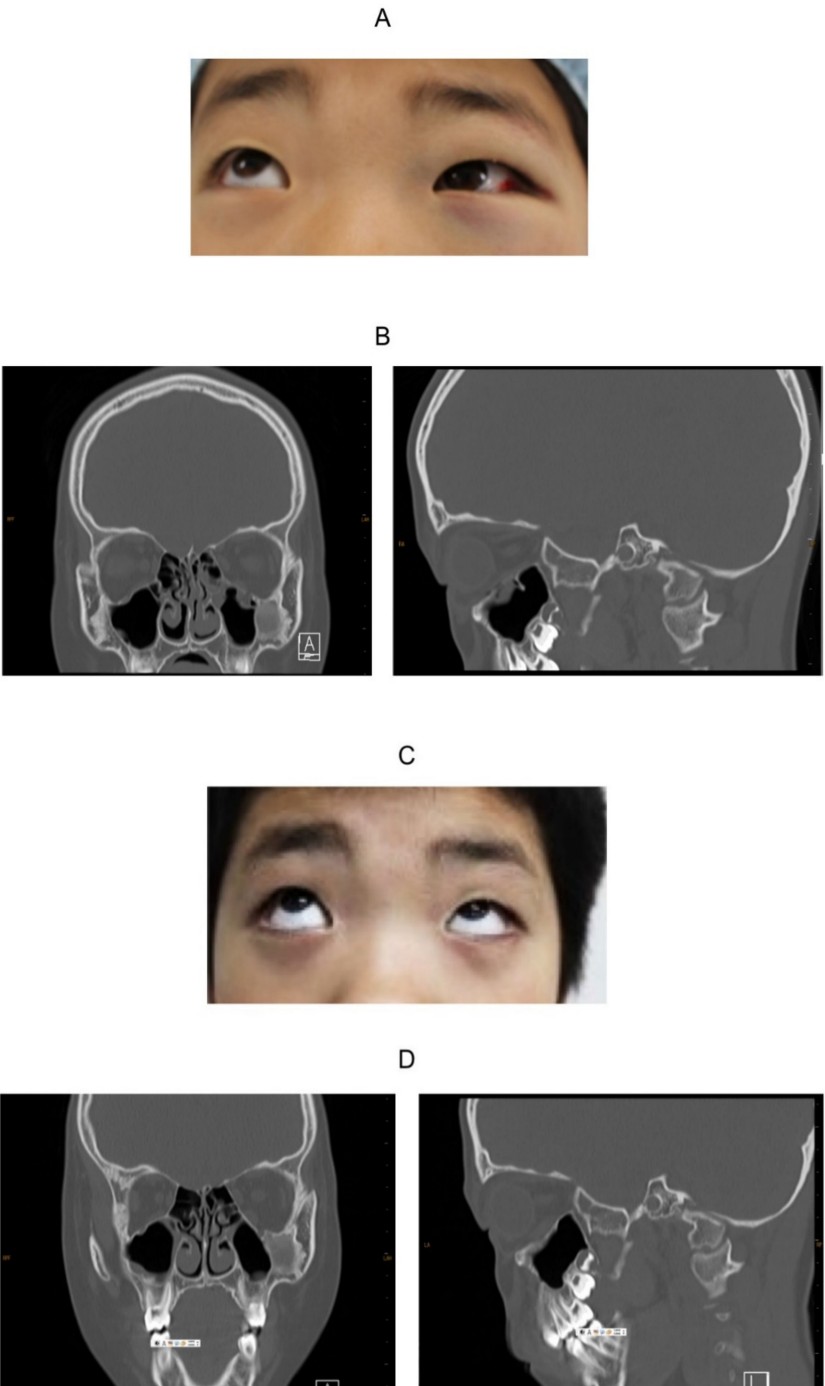

**Figure 6.** Findings of Case 10. (**A**) The left locked eye looking upward. (**B**) CT image before reconstruction. The left and the right panels respectively portray coronal and sagittal images. The orbital soft tissue is dislocated into the left maxillary sinus. The soft tissue density of the left maxillary bone is fibrous dysplasia, not related to the fracture. (**C**) The eye movement was improved completely through the operation. (**D**) The CT image after reconstruction. The left and right panels respectively portray coronal and sagittal images. The left orbital floor is repaired without implant.

## 4. Discussion

Reconstruction of orbital floor fracture has remained controversial through the decades in terms of its indication, approaches, and timing of operation. Regarding indication of reconstruction, prolonged oculocardiac reflex, white-eyed BOF, enophthalmos, diplopia in eye movement, breakdown over 50% of the orbital floor, and entrapment of orbital soft tissues were described in an earlier report [3].

Common guidelines for timing of the BOF or TF operation have not been defined [4]. Reportedly, BOF without constriction of the orbital soft tissues does not always require early reconstruction in 2 weeks [5]. However, wound healing with fibrosis around the fracture can be expected to proceed during observation, leading to difficulty in reconstruction. Early operation of BOF within 2 weeks reportedly reduces complications, including diplopia, enophthalmos, and infraorbital nerve dysfunction [6]. Consequently, we suggest that operations should be performed for BOF patients with operative indication within 2 weeks after injury [7] or within 4 weeks at the latest, based on our own experience. By contrast, most TF in children should be reconstructed within 24 h because entrapment of the inferior rectus muscle can engender cicatrization and unrecoverable diplopia [8,9]. Operations for our patients were conducted around 24 h after injury (Case 9: 6 h, Case 10: 26 h, Case 11: 7 h), leading to complete recovery. Even if the entrapment is unclear on CT scan images, TF with locked eye movement, oculocardiac reflex, or vomiting should be treated promptly with operation, especially in children [10,11].

Surgical procedures for orbital floor fractures have routinely involved a transantral, transorbital, or endonasal endoscopic approach, or some combination of them [12–15]. Although we chose only the transantral approach before 2008, diplopia persisted in some patients through single approach reconstruction. In light of a report suggesting that incarceration of orbital tissue cannot be released completely by the single transantral approach [16], since 2009 we have adopted a combination of transantral and transorbital approaches: a combined approach [12]. In the transantral approach, the orbital tissues can be lifted upward through the maxillary sinus using an endoscope. Using a transorbital approach, the orbital floor can be approached to pull orbital tissues upward via subciliary incision. The single transorbital approach is useful for fractures in the anterior part of the orbital floor, but it has less benefit for fractures in the posterior part [12]. In contrast, the single transantral approach can serve surgeons with easy and precise manipulation for the posterior part of the orbital floor. The approach combining them overcomes these shortcomings, with mutually complementary aspects, with no postoperative esthetic complications [12,14]. Our results demonstrated that postoperative diplopia was present in 4.8% of the patients. No esthetic complication was found, suggesting possible superiority of a combined approach to single transorbital reconstruction in which incidence of postoperative diplopia was 9.1–42.5% [6,17]. Recently, Chai et al. recommended an absorbable implant using a personalized 3D printing technique for pediatric TF [18]. A combined approach can yield results comparable to theirs (postoperative diplopia in 5.6%). Although a combined approach takes longer than a single approach, our operation time (median = 149 min) is tolerable. Combined approaches, including transantral (Caldwell–Luc) approach, are aggressive for younger people, especially for children. However, returning bone of the maxillary anterior wall and careful closing of the wound lead to recovery of the maxillary sinus with aeration, avoiding complications such as sinusitis.

For a transorbital approach, skin or conjunctiva incision is needed. Conjunctiva incision on the lower eyelid is reported as superior to skin incision, including subtarsal or subciliary incision in avoiding esthetic problems such as a visible scar or lower eyelid malposition [19]. However, orbital fat tissue can disturb the clear visual field under a microscope through conjunctiva incision. For broad surgical exposure of the orbital floor, subciliary incision is recommended [19]. With careful and gentle manipulation for eyelids, we have experienced no esthetic complication through subciliary incision in any child or adult patient, as described in an earlier report [13]. Even conjunctiva incision is not always an ideal procedure, reportedly causing complications such as entropion or scleral

show [14]. The operator should select approach options depending on their experience and preference, although one can infer that subciliary incision is adaptable for children during their growth and development.

## 5. Conclusions

Operations for BOF and TF with the single transorbital approach are frequently performed by maxillofacial surgeons or ophthalmologists. A combined approach using endoscopy requires a longer operation time than a single transorbital approach. However, a combined approach can yield the benefit of reducing the incidence of postoperative diplopia, which is usually permanent, thereby impairing a patient's QOL for a long time. A combined approach should be regarded as an option for TF and certainly for BOF in children. After otorhinolaryngologists with abundant experience in using an endoscope through paranasal sinuses learn the transorbital procedure, the combined approach represents a promising and feasible technique to achieve reconstruction.

**Author Contributions:** Conceptualization, N.N. and Y.I. (Yumi Ito); writing—review and editing, N.N., M.O. and T.T.; operation, N.N., Y.K. (Yukinori Kato), Y.K. (Yukihiro Kimura), Y.I. (Yoshimasa Imoto), and K.O., supervision, S.F. All authors have read and agreed to the published version of the manuscript.

**Funding:** This research received no external funding.

**Institutional Review Board Statement:** The study was conducted according to the guidelines of the Declaration of Helsinki, and approved by the ethical board of the University of Fukui (protocol code 20210002C approval 7 May 2021).

**Informed Consent Statement:** Written informed consent has been obtained from the patients to publish this paper.

**Data Availability Statement:** The data presented in this study are available on request from the corresponding author. The data are not publicly available due to restricting privacy.

**Acknowledgments:** We thank H. Yamamoto for a critical review of this work. We are also grateful to H. Tsuchiya and M. Nakano for excellent technical assistance.

**Conflicts of Interest:** The authors report that they have no conflicts of interest. The authors alone are responsible for the contents and composition of the article.

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
