# Peer review of "Combined Subciliary/Transantral Approach for Reconstruction of Orbital Floor Fracture"

_2504-463X, doi:10.3390/ohbm2030007_

Round 1

Reviewer 1 Report

The double approach is not justified. The number of patients treated is too low and is not scientifically valid.

Author Response

Response to reviewer #1's comments

Thank you very much for your detailed review of our report.

Comment

The double approach is not justified. The number of patients treated is too low and is not scientifically valid.

response

According to the reviewer’s comment, we added clinical data of patients who received combined approach reconstruction for BOF or TF in our department between August 2009 and March 2021 (Table 1). The revised manuscript includes analysis of pre-operative and post-operative symptoms of 21 patients, demonstrating the usefulness of a combined approach for BOF and TF reconstruction.

Reviewer 2 Report

Strength of the paper:

Trapdoor orbital fractures (TF) are common in children. This particular type of orbital blowout fracture can involve inferior wall, medial wall, or both, although inferior orbital wall is most often affected. For the particular symptomatology characterized by nausea, vomiting and bradycardia derived from oculo-cardiac reflex, these fractures request an immediately treatment. The Authors present an interesting paper with two cases of TF treated by combined trans-orbital subciliary incision and trans-antral like Caldwell-Luc approach. The cases are well described and the obtained results are optimal. Good iconography supports the data. In general, the article is well written and in correct English.

Weakness of the paper:

  • The literature used to support their thesis is obsolete [12,13], in particular as regards the transantral approach. Recent literature accepts the use of this approach as revision surgery in adults. In children's trapdoor fractures, since there is an elastic return of the fractured fragments, a Caldwell-luc type approach is too aggressive, exposing the child to sinusitis and other complications and longing surgery time.
  • Although the Authors explained that the transcutaneous approach was chosen by the surgeon for preference and experience, recent literature supports a transconjunctival approach in children to avoid aesthetic complications. Furthermore, several authors recommend the use of 3d-printing software to ensure greater accuracy and safety of treatment [Chai, G., Zhang, D., Hua, W., Yin, J., Jin, Y., & Chen, M. (2021). Theoretical model of pediatric orbital trapdoor fractures and provisional personalized 3D printing-assisted surgical solution. Bioactive Materials, 6(2), 559–567]

Author Response

Response to reviewer #2's comments

Thank you very much for your valuable comments and suggestions.

Comment

Weakness of the paper:

  • The literature used to support their thesis is obsolete [12,13], in particular as regards the transantral approach. Recent literature accepts the use of this approach as revision surgery in adults. In children's trapdoor fractures, since there is an elastic return of the fractured fragments, a Caldwell-luc type approach is too aggressive, exposing the child to sinusitis and other complications and longing surgery time.

response

Thank you very much for your valuable suggestion. As the reviewer’s comment, the combined approach is not the state-of-the-art technique. However, it has potential benefits for reconstruction of BOF or TF. We cited additional recent papers describing the use of a combined approach [14,15].Furthermore, we added clinical data of patients who received combined approach reconstruction for BOF or TF in our department between August 2009 and March 2021 (Table 1).Indeed application of the Caldwell–Luc approach for younger patients seems to be aggressive, but we have found no complication among the patient data examined for this study (Table 1). The patients’ maxillary sinuses were healed with aeration with return of anterior wall bone of maxillary sinus into the original position (additionally described in case 9 presentation). The surgery time was longer in the combined approach. Although the median duration of the combined approach was 149 min, the complete recovery ratio of diplopia was 95.2%. We hope that the good effectiveness can justify the longer operation time associated with the combined approach.

Comment

  • Although the Authors explained that the transcutaneous approach was chosen by the surgeon for preference and experience, recent literature supports a transconjunctival approach in children to avoid aesthetic complications. Furthermore, several authors recommend the use of 3d-printing software to ensure greater accuracy and safety of treatment [Chai, G., Zhang, D., Hua, W., Yin, J., Jin, Y., & Chen, M. (2021). Theoretical model of pediatric orbital trapdoor fractures and provisional personalized 3D printing-assisted surgical solution. Bioactive Materials, 6(2), 559–567]

response

Thank you very much for your detailed review. As the reviewer has described, some operators favor conjunctiva incision. We adopt subciliary incision for wide surgical exposure of the orbital floor [19]. With careful operation, we have found no complication of lower eyelid after subciliary incision. Severe complications including entropion or scleral show were reported even in conjunctiva incision [14]. We think that the choice of subciliary or conjunctiva incision is dependent mainly on the operator’s experience and expertise.

We agree with the reviewer’s comment. Absorbable implants using 3D printing technique is an ideal procedure with a high success rate [18]. We demonstrated that a combined approach was comparable to results obtained using 3D-printing technique for postoperative permanent diplopia. Our results indicate the possibility of successful reconstruction without the use of the cutting-edge 3D-printing technique, which is still difficult to use in many areas of the world.

Reviewer 3 Report

Dear authors, an exciting topic, thanks for the submission. Even though the topic seems interesting and relevant, I see significant unanswered questions.

In the reconstruction of the orbital floor, there is now a very broad knowledge base, although some areas are still very controversial. This certainly includes pediatric orbital fractures, but there is a consensus that rapid therapy is required if entrapment occurs.

The present work is a 2-case report, and this should be mentioned in the title. In the literature search, it appears that significant publications were not included.

In addition, it is not sufficiently explained why 2 access routes are chosen when there are very elegant, minimally invasive therapy options also in children.

The reappraisal in the discussion pro and con is missing. In addition, in my opinion, essential examination parameters, diagnostic measures, details of the operation (how long does the operation take here), follow-up, etc. are missing. Use of autologous or alloplastic material for large defects etc ...

The technique has been used since 2009 and there are only 2 patients?  Why is the case data from 12 years not reviewed and presented here? Overall an exciting topic, but the article still needs some effort to be clinically relevant.

Author Response

Response to reviewer #3's comments

Thank you very much for your valuable comments and detailed suggestions.

Comment

Dear authors, an exciting topic, thanks for the submission. Even though the topic seems interesting and relevant, I see significant unanswered questions.

In the reconstruction of the orbital floor, there is now a very broad knowledge base, although some areas are still very controversial. This certainly includes pediatric orbital fractures, but there is a consensus that rapid therapy is required if entrapment occurs.

The present work is a 2-case report, and this should be mentioned in the title. In the literature search, it appears that significant publications were not included.

response

Thank you very much for your suggestion. As the reviewer has commented, we added clinical data of patients who received combined approach reconstruction for BOF or TF at our department between August 2009 and March 2021 (Table 1). We changed the title of our article to represent the revised manuscript.

 Also, it is not sufficiently explained why 2 access routes are chosen when there are very elegant, minimally invasive therapy options also in children.

response

We adopted a combined approach for TF in children because we know that its success rate is high for improvement of diplopia for some cases. Postoperative diplopia is usually permanent; it impairs the QOL of the patients.

The reappraisal in the discussion pro and con is missing. In addition, in my opinion, essential examination parameters, diagnostic measures, details of the operation (how long does the operation take here), follow-up, etc. are missing. Use of autologous or alloplastic material for large defects etc

response

Thank you very much for your valuable suggestion.We presented detailed clinical findings of the patients and outcomes of the operation (Table 1and Results).Success rates of combined approaches for diplopia and longer operation times represent benefits and shortcomings of the combined approach. We described that point in the Discussionsection.

The technique has been used since 2009 and there are only 2 patients?  Why is the case data from 12 years not reviewed and presented here? Overall an exciting topic, but the article still needs some effort to be clinically relevant.

response

According to the reviewer’s comment, we added clinical data of 21 patients who received combined approach reconstruction for BOF or TF. Patients with multiple fractures of maxillofacial bone or insufficient follow-up were excluded. Materials and Methods, Results, and Table 1 were augmented or improved.

Round 2

Reviewer 3 Report

The authors have significantly improved the manuscript. Linguistic / grammatical revision is still advisable. It is also questionable how the authors come to the conclusion that a combined intervention (with increased morbidity) should be superior to approaches such as the proven transconjunctival approach. The fractures shown could have been treated without any problems using this approach, presumably with a significantly shorter operation time. Here, the authors state an operating time of at least 2 hours - a relatively long time compared to classic orbital floor reconstructions by cranio-maxillofacial surgeons. The apparent "superiority" should definitely be explained in the discussion and the conclusion should be adjusted.

The authors have significantly improved the manuscript. Linguistic / grammatical revision is still advisable. It is also questionable how the authors come to the conclusion that a combined intervention (with increased morbidity) should be superior to approaches such as the proven transconjunctival approach. The fractures shown could have been treated without any problems using this approach, presumably with a significantly shorter operation time. Here, the authors state an operating time of at least 2 hours - a relatively long time compared to classic orbital floor reconstructions by cranio-maxillofacial surgeons. The apparent "superiority" should definitely be explained in the discussion and the conclusion should be adjusted.

"Operations for BOF and TF with single transorbital approach are frequently performed by maxillofacial surgeons or ophthalmologists. A combined approach using endoscopy is superior to single approaches. It should be regarded as an option for TF and certainly for BOF in children. After otorhinolaryngologists with abundant experience in using an endoscope through paranasal sinuses learn the transorbital procedure, the combined approach represents a promising and feasible technique to achieve reconstruction."

Author Response

Response to reviewer #3's comments

Thank youvery much for your valuable comments and detailed suggestions.

Comment

The authors have significantly improved the manuscript. Linguistic / grammatical revision is still advisable. It is also questionable how the authors come to the conclusion that a combined intervention (with increased morbidity) should be superior to approaches such as the proven transconjunctival approach. The fractures shown could have been treated without any problems using this approach, presumably with a significantly shorter operation time. Here, the authors state an operating time of at least 2 hours - a relatively long time compared to classic orbital floor reconstructions by cranio-maxillofacial surgeons. The apparent "superiority" should definitely be explained in the discussion and the conclusion should be adjusted.

"Operations for BOF and TF with single trans orbital approach are frequently performed by maxillofacial surgeons or ophthalmologists. A combined approach using endoscopy is superior to single approaches. It should be regarded as an option for TF and certainly for BOF in children. After otorhinolaryngologists with abundant experience in using an endoscope through paranasal sinuses learn the trans orbital procedure, the combined approach represents a promising and feasible technique to achieve reconstruction."

Response

Thank you very much for your thoughtful advice. As the reviewer has commented, we revised the Conclusionsection. Indeed, a combined approach is inferior to a single trans-orbital approach in terms of shorter operation time. However, a combined approach demonstrated less post-operative diplopia (4.8%) than described in earlier reports about a single trans-orbital approach (9.1–42.5%)[6, 17]. Post-operative diplopia is usually permanent, impairing the patients’ QOL for a long time. Therefore, we conclude that a combined approach has a potential advantage to reduce post-operative diplopia.